# Automated Identification of Cutaneous Leishmaniasis Lesions Using Deep-Learning-Based Artificial Intelligence

**DOI:** 10.3390/biomedicines12010012

**Published:** 2023-12-20

**Authors:** José Fabrício de Carvalho Leal, Daniel Holanda Barroso, Natália Santos Trindade, Vinícius Lima de Miranda, Rodrigo Gurgel-Gonçalves

**Affiliations:** 1Graduate Program in Tropical Medicine, Center for Tropical Medicine, Faculty of Medicine, University of Brasília–UnB, Brasília 70904-970, Brazil; fabriciolealc29@gmail.com; 2Laboratory of Medical Parasitology and Vector Biology, Faculty of Medicine, University of Brasília–UnB, Brasília 70904-970, Brazil; nataliatrindadeunb@gmail.com (N.S.T.); viniciuslimabio@gmail.com (V.L.d.M.); 3Postgraduate Program in Medical Sciences, Faculty of Medicine, University of Brasília–UnB, Brasília 70904-970, Brazil; danielhbarroso@unb.br

**Keywords:** dermatology, leishmaniasis, diagnosis, AlexNet, machine learning, pictures

## Abstract

The polymorphism of cutaneous leishmaniasis (CL) complicates diagnosis in health care services because lesions may be confused with other dermatoses such as sporotrichosis, paracocidiocomycosis, and venous insufficiency. Automated identification of skin diseases based on deep learning (DL) has been applied to assist diagnosis. In this study, we evaluated the performance of AlexNet, a DL algorithm, to identify pictures of CL lesions in patients from Midwest Brazil. We used a set of 2458 pictures (up to 10 of each lesion) obtained from patients treated between 2015 and 2022 in the Leishmaniasis Clinic at the University Hospital of Brasilia. We divided the picture database into training (80%), internal validation (10%), and testing sets (10%), and trained and tested AlexNet to identify pictures of CL lesions. We performed three simulations and trained AlexNet to differentiate CL from 26 other dermatoses (e.g., chromomycosis, ecthyma, venous insufficiency). We obtained an average accuracy of 95.04% (Confidence Interval 95%: 93.81–96.04), indicating an excellent performance of AlexNet in identifying pictures of CL lesions. We conclude that automated CL identification using AlexNet has the potential to assist clinicians in diagnosing skin lesions. These results contribute to the development of a mobile application to assist in the diagnosis of CL in health care services.

## 1. Introduction

Leishmaniasis is an infectious, neglected, and emerging tropical disease caused by *Leishmania* parasites transmitted through the bite of infected sandflies. The clinical presentation of leishmaniasis varies depending on the *Leishmania* species and the interaction with the host. Cutaneous leishmaniasis (CL) is an ancient disease, described in Egyptian papyri from 2000 BC. In the 19th century, it was believed that CL was caused by fungi. Moreover, there have been descriptions of skin ulcers in the Andes and Central America, which have also been mistakenly identified as leprosy. These historical reports demonstrate the difficulty in diagnosing CL [1,2].

CL is caused mainly by *L. mexicana*, *L. amazonensis*, *L. braziliensis*, *L. panamensis*, *L. tropica*, *L. major*, and *L. aethiopica*. An estimated 350 million people live in regions with active *Leishmania* transmission across five continents in approximately 100 countries [1]. The annual incidence rate of leishmaniasis ranges from 0.7 to 1 million cases [3]. Most cases of CL (85%) occur in ten countries (Afghanistan, Algeria, Brazil, Ethiopia, Iran, Peru, Sudan, Costa Rica, Colombia, and Syria) [3,4,5].

The most prevalent clinical form of leishmaniasis is CL, causing single or multiple exudative painless lesions with raised borders [3]. The lesions develop over weeks to months when the papule enlarges to a nodule that ulcerates slowly. Although less common, leishmaniasis can also manifest as verrucous, nodular, tuberous lesions, among other forms. CL is usually manifested as a self-limited ulcer that can heal spontaneously over 3–18 months but cases progress to more severe manifestations depending on parasite species [3,6,7,8,9]. The polymorphism of CL complicates diagnosis in health care services because lesions may be confused with other dermatoses caused by fungi (sporotrichosis, paracoccidioidomycosis, chromomycosis), bacteria (leprosy, pyodermite, ecthyma) and non-infectious diseases (venous insufficiency, skin cancer). Differential diagnosis is important since similar diseases are common in CL endemic regions. The diagnosis depends on a detailed clinical assessment considering signs and symptoms, epidemiological data, and laboratory tests, resulting in delayed treatment. There is no single reference test for CL and a combination of different methods is recommended [3,10,11,12,13,14]. For this reason, the development of new diagnostic methods for CL is of fundamental importance.

Recent advances in artificial intelligence (AI) have revolutionized medicine, with machine learning (ML) algorithms and their identification capacity having an important application to help diagnosis [15,16,17,18]. Furthermore, dermatology has benefited from novel diagnostic tools. Computational algorithms have been applied to help dermatologists diagnose skin diseases, such as melanomas, improving accuracy and early detection [19,20,21,22,23,24,25,26]. However, there are few studies applying ML to diagnose dermatoses caused by infectious agents such as *Leishmania*. Bamorovat et al. [18], presented a new diagnostic and prognostic method for classifying CL including responsive and non-responsive patients, showing the potential for automatic identification of leishmaniasis using ML algorithms. Furthermore, Noureldeen et al. [27], showed a new diagnostic method of CL detection and classification with an ML model. A new step in this area is to analyze pictures of CL compared to other skin diseases caused by fungal and bacterial infections, and non-infectious diseases, and applying pre-trained neural networks such as AlexNet to aid in the diagnosis of CL.

Deep learning (DL) techniques have potential to benefit the diagnosis in dermatology through the analysis of pictures of skin lesions, as they extract patterns from pictures and make predictions from large databases. Given this, the application of the technique can be an important step to support an accurate diagnosis, facilitating and assisting clinical decision making [28,29,30]. Therefore, the present study aimed to evaluate the performance of the DL algorithm (AlexNet) in identifying pictures of CL lesions from patients in Midwest Brazil.

## 2. Materials and Methods

### 2.1. Study Design and Picture Database

We obtained pictures of skin lesions from patients attending from 2015 to 2022 in the Leishmaniasis Clinic at the University Hospital of Brasilia, a referral service for leishmaniasis diagnosis and treatment in the Brazilian mid-western region. In the first consultation, we photographed the skin lesions (up to 10 pictures of each patient lesion) with a Sony alpha 7 R III-ICLE-R7M2 camera (Sony Corp, Tokyo, Japan) and Sony FE 24–105 mm or F4 G OSS or Sony FE 35 mm f/2.8 lenses (Sony Corp, Tokyo, Japan). Moreover, skin biopsies and mucocutaneous smears were performed to confirm the diagnosis [31,32,33].

We classified patients according to the composite reference standard following the operational classification of the Brazilian Ministry of Health [34]. The diagnosis for CL was confirmed after the observation of compatible clinical symptoms, with two other non-parasitological positive tests (indirect immunofluorescence, Montenegro intradermal reaction) or one parasitological positive test (qPCR, culture, histopathology, or smear with amastigotes) (Figure 1). After clinical evaluation by the dermatologist, the lesions were subjected to additional tests to confirm leishmaniasis and other differential diagnoses. The color (erythematous, hypo or hyperpigmented), body location (upper limbs, lower limbs, face, or trunk), diameter and type of lesions (macule, papule, plaque, nodule, ulcer, vesicle, or blister) were recorded. Therefore, the final diagnosis was only established after laboratory results were available and allowed dermatologists to communicate the diagnosis and begin treatment.

We classified patients according to the composite reference standard [34]. Therefore, we included all patients with lesions that had diagnoses for CL and other dermatoses. We excluded patients if their skin lesions were located at mucous membranes, intimate body areas and the scalp, due to the difficulty of taking quality pictures without trichotomy. Patients who did not complete the diagnostic assessment were also excluded. The polymorphism of CL means that differential diagnosis with other diseases must always be considered. Therefore, patients diagnosed with other types of skin diseases also participated in this study (Table 1).

We obtained up to 10 pictures of each patient lesion: panoramic, close-up, lateral and edges. We identified pictures with an alphanumeric code and saved on these an encrypted disk to protect the participants’ identity. Therefore, we created an image database with each type of skin lesion. We resized the pictures to improve visual quality, aiming to remove unwanted areas and highlight the region of interest. We cropped the pictures of facial lesions to avoid patient identification. Therefore, we compiled a database containing 1787 pictures of CL and 671 of other dermatoses (Table 1), resulting in a total of 2458 high-resolution pictures used to train and test the AlexNet algorithm (Figure 2).

### 2.2. Algorithm: Training and Testing

We used a DL algorithm (AlexNet) to identify skin lesion pictures. AlexNet is a Convolutional Neural Network which has 60 million parameters and 650,000 neurons. AlexNet was pre-trained on 1.2 million images of varying resolution and representing 1000 classes (i.e., objects, animals, plants) from the ImageNet database [35]. AlexNet has been successfully used for automated identification of insect vectors [36,37] and disease diagnosis based on images [38,39].

AlexNet operates according to some fundamental components and principles: (1) it has five convolutional layers, responsible for extracting relevant features from the input images, which detect specific patterns, such as edges, textures and shapes; (2) it has max pooling layers that reduce the dimensionality of the image representation, preserving significant characteristics and reducing computational processing; (3) it uses the ReLU (Rectified Linear Unit) function in the convolutional layers to add non-linearity to the network, allowing it to learn more complexities in the data; (4) it uses the Local Response Normalization technique to normalize the outputs of the convolutional layers, promoting competition between neurons and increasing the generalization capacity; (5) in addition to the convolutional and pooling layers, it has three fully connected layers that combine the learned features to perform the final classification; (6) it employs the Dropout technique to combat overfitting, randomly deactivating some neurons during training, thus preventing the network from becoming too dependent on some neurons; and (7) it uses the Softmax function to convert the network outputs into probabilities so that the highest probability is chosen as the final prediction [35].

We cropped all pictures into a square format (227 × 227 pixels) according to the input patterns required by the AlexNet. All pictures were saved in the red-green-blue (RGB) format at a 72 dpi resolution. We divided the picture database into training (80%), internal validation (10%) and testing sets (10%), and trained and tested AlexNet to identify skin lesion pictures. In the learning process, AlexNet randomly chose the training, internal validation, and test pictures. For training, we defined the maximum number of epochs for identifying CL lesions (Figure 3) versus other dermatoses (Figure 4) as 100. Furthermore, network analyzes were performed in the MATLAB computational environment (www.mathworks.com, accessed on 10 October 2023).

### 2.3. Data Analysis

We ran three simulations in AlexNet. First, we trained the algorithm to learn to distinguish CL lesions from other dermatoses. Next, we evaluated the performance of the algorithm using confusion matrices, analyzing the number of hits and identification errors, as well as the average network accuracy, sensitivity, and specificity. We used the following equations to calculate the ML performance indicators:Accuracy = (VP + VN)/(VP + FP + VN + FN)
Sensitivity = VP/(VP + FN)
Specificity = VN/(VN + FP)

VP = True positives;

VN = True negatives;

FP = False positives;

FN = False negatives.

We calculated frequencies and proportions with confidence intervals (Wilson binomial with a 95% score) using the ‘Hmisc’ package in the computational software R 4.2.1, together with the RStudio 2023.03.1.446 interface [40,41,42,43]. Therefore, the results were presented through descriptive tables and illustrative graphs.

### 2.4. Ethical Consideration

This study received approval from the Ethics and Research Committee of the Faculty of Medicine of the University of Brasília, CAAE: 68696323.5.0000.5558 and number: 035700/2023.

## 3. Results

### 3.1. Training

The results of the three simulations demonstrated significant progress, in which the validation accuracy achieved good results. The algorithm’s training process was completed within 100 epochs and iterations. We observed that AlexNet learns from new data, because as the epochs progress, data accuracy increases and, consequently, the losses decrease (Appendix A).

### 3.2. AlexNet Performance

Simulation 1 showed 465 correct identifications and 26 errors, with an accuracy of 94.70% (95% CI 92.35–96.36). Seven pictures of CL lesions were confused with lesions from other types of dermatoses, while nineteen lesions from other types of diseases were confused with CL (Figure 5A).

Simulation 2 showed 469 correct identifications and 22 errors, with an excellent accuracy of 95.51% (95% CI 93.30–97.02). Of the wrong pictures, three were of CL lesions that were confused with lesions from other types of dermatoses, and nineteen were of lesions from other types of dermatoses that were confused with CL (Figure 5B). In simulation 3, we obtained an accuracy of 94.90%, represented by 466 correct identifications and 25 errors (95% CI 92.59–96.52). In this simulation, three pictures of CL lesions were confused with other dermatoses. In contrast to the other two simulations, 22 pictures of other types of dermatoses were confused with CL (Figure 5C).

Among the 60 pictures of lesions from other dermatoses that were confused with CL in these three simulations, those with the highest percentage of errors were leprosy (16.7%), ecthyma (11.6%), BCC (10.0%), erysipelas (8.3%) and lichen planus (8.3%) (Figure 6).

During the three simulations, we achieved an average validation accuracy of 95.04% (95% CI 93.81–96.04), indicating excellent performance of the trained network for the task of identifying CL lesions. Figure 7 shows the accuracy, sensitivity, and specificity of each of the simulations with their respective confidence intervals.

## 4. Discussion

Our results show an excellent performance of AlexNet in identifying pictures of CL lesions. These results pave the way to develop an AI system based on ML that can facilitate the diagnosis of CL lesions, assist clinicians in the differential diagnosis from other skin diseases, promote faster diagnostic, timely treatment, and consequently help improve the quality of life of patients diagnosed with CL.

Our database was built based on a cohort of patients treated at the Leishmaniasis Clinic of the University Hospital of Brasília over 8 years. All skin lesions were pictured during the patients’ first contact. However, the pictures were only added to the database after a conclusive diagnosis of the skin disease [27,28,29] and they were separated into folders according to the disease. Therefore, the diagnosis followed the composite reference standard protocol established by the Brazilian Ministry of Health [34]. These data show that our database is reliable and based on a well-established diagnostic protocol to ensure the identification of the CL lesions included in this study, which were used for automated identification using AlexNet.

Technologies, medical interventions, and diagnostic tests are not perfect and must be evaluated before introduction into health care services. Therefore, diagnostic accuracy studies are important to discriminate between infected and non-infected individuals [44]. This ability can be quantified through measures such as sensitivity and specificity. Furthermore, the main objectives of diagnostic tests and other forms of testing should be to prevent early death, decrease suffering, and restore functional health. In this way, new tools that improve the diagnosis, support patients with the disease, and help clinicians to detect sick individuals should be developed [45,46].

AI refers to the ability of a machine to simulate human intelligence [47]. AI introduces new concepts and solutions to address complex challenges [48]. ML involves algorithms and statistical models designed to learn from examples and observations. In this way, ML can recognize and infer patterns, enabling it to execute tasks without requiring explicit instructions from a human operator [47,49]. ML includes three types of algorithms: (i) unsupervised, which finds patterns without external guidance; (ii) supervised, which requires human input to help the machine learn; and (iii) reinforcement learning, which uses rewards and punishments to create a specific operational strategy [48]. ML-based techniques have proven successful in various fields, including biomedical and medical applications [50]. Neural networks are a common type of supervised ML that has been widely used in the medical field.

During training, the data flow through an iterative network where weights are constantly adjusted, enabling the neural network to enhance its classification capacity [47]. A visual representation can be found in Appendix A. DL with multiple layers of neurons has emerged as the leading AI technique for processing complex image data. DL utilizes artificial neural networks to map image inputs and diagnostic outputs without the explicit involvement of human engineering. It should be noted that convolutional neural networks (CNNs) are a particular type of DL that have been proven successful in classifying image data. The use of CNNs for image classification has become increasingly popular due to advancements in computer-based methods like AlexNet [47,51], which utilize transfer learning from pre-trained networks to learn from images in datasets [34]. Classifying data with CNNs is an easy, cost-effective, and computationally efficient process. In medicine, CNNs have primarily been used for visual diagnosis in dermatology, radiology, and pathology [47].

There are over 2000 skin conditions that impact one-third of the global population, constituting a significant health care burden [52]. Dermatology has the potential to benefit from the use of DL to enhance patient care. Thus, the combination of AI and hardware-based methods can boost dermatologists’ abilities and is essential in improving access to and the quality of dermatological care. This provides real-time diagnostic support during clinical consultations [51]. Choy et al. [52] analyzed the most common skin conditions, including acne, psoriasis, eczema, rosacea, and urticaria. The DL algorithms have an estimated accuracy rate of around 90% for diagnosing these diseases [52]. Srinivasu et al. [21] developed a process utilizing various DL algorithms to classify non-infectious skin diseases. Their study showed that MobileNet V2 had greater accuracy on lightweight computing devices. Goceri [23] used MobileNet V2 in a mobile phone application with a user-friendly interface and demonstrated that the proposed method accurately diagnosed dermatological diseases with 94.76% accuracy. AlSuwaidan [53] evaluated the performance of six CNNs (VGG16, EfficientNet, InceptionV3, MobileNet, NasNet and ResNet50) for the three most common dermatological conditions (eczema, atopic dermatitis, and psoriasis) in the Middle East, and MobileNet had the highest accuracy (95.7%). Li et al. [54] also showed the performance of different CNNs for the diagnosis of actinic keratosis with an accuracy of about 92%. Bisla et al. [55] employed DL for melanoma classification, displaying superior performance compared to common baseline methods. Finally, Noureldeen et al. [27] employed the YOLOv5 model for identifying CL and obtained a mean precision of 70% in pinpointing skin lesions. Thieme et al. [56]. developed a CNN to identify the characteristics of skin lesions caused by monkey pox, achieving a sensitivity of approximately 90%. These data suggest that DL mobile systems may assist clinicians in accurately diagnosing skin lesions. Additionally, AI systems could be a way to speed up the diagnostic process and reduce the high daily workload, and DL approaches have provided highly satisfactory results in image analysis to achieve this goal [57].

Other studies have used DL models for dermatological diagnosis and have performed equally or better than dermatologists, e.g., Han et al. [19]. Currently, there are no studies that have analyzed pictures of CL lesions in comparison with other skin diseases caused by fungal infections, bacterial infections, and non-infectious diseases by applying a pre-trained neural network such as AlexNet. In this study, we showed that the DL network has an average accuracy of 95.04% for identifying pictures of CL lesions from other skin diseases. Therefore, the performance of the algorithm used in our study will help to improve the diagnosis of CL based on pictures of skin lesions.

We observed that leprosy, ecthyma and basal cell carcinoma (BCC) were frequently confused with CL when using AlexNet. CL lesions can have diverse clinical manifestations, with atypical morphologies, which may be associated with several factors, such as pathogenicity, virulence, and host immunity [14]. Differential diagnosis still represents a challenge, as other diseases can manifest in a similar way to CL, such as BCC, leprosy, ecthyma and malignant neoplasm. Furthermore, other skin diseases that could be confused with CL include chromomycosis, blastomycosis, cutaneous tuberculosis, squamous cell carcinoma, erysipelas, herpes zoster, cutaneous lymphoma, tertiary syphilis, leprosy, paracocciomycosis, lupus, granulomatous rosacea, and persistent reactions to arthropod bites, among others [10,11,12,56]. Therefore, CL is called “the great imitator”, as it can resemble several types of dermatoses [14]. It is well established that when multiple comparisons are performed, an increase in the amount of type 1 errors is expected [58]. By the same logic, when comparing multiple categories, it is expected that only by chance will some categories exhibit a higher rate of false positives than others. The presence of various diseases in the “other dermatoses” category results in few images of each disease type, which can complicate the training of the AI algorithm for leprosy, ecthyma, and BCC, increasing the likelihood that a particular disease will have a high false-positive rate.

Although our study demonstrates the potential to differentiate CL from other skin lesions via DL, it also has limitations. The Leishmaniasis Clinic of the University Hospital of Brasilia, despite being a reference center for CL, may have a different patient profile than primary care patients, which may affect the accuracy of the test [59]. Therefore, the results may not be completely applicable to the general population, as the profile of those served in basic care is different. To overcome this situation, sampling could be carried out that includes primary care patients. Another limitation is the number of pictures for the two groups used as a database, especially for other skin diseases, as there is a lower proportion of patients referred to the University Hospital of Brasilia and diagnosed with other dermatoses during the years the database was constructed, since the clinic is a reference for CL diagnosis. However, collaborations with other dermatology centers to increase the database of other dermatoses may be a possible solution. It is important to expand the database to include CL lesions and other skin diseases caused by various *Leishmania* species found in other countries, as our study indicates that the lesions are predominantly caused by *L. braziliensis*, and in some cases, by *L. amazonensis* and *L. guyanensis* [60,61]. It is noteworthy that Old World CL lesions differ from New World lesions, which may progress to hyperkeratotic plaques. When caused by *L. tropica* or *L. major*, the wounds can self-heal within a year and may leave scars. However, if caused by *L. aethiopica*, healing may take years and result in severe mucocutaneous oronasal leishmaniasis. New World CL lesions caused by *L. mexicana* are less severe and heal faster compared to lesions caused by the *Leishmania* (*Viannia*) species, which result in more severe forms of ulcerative and mucocutaneous leishmaniasis [3]. We anticipate that the polymorphism of skin lesion pictures in the database could enable the AI algorithm to distinguish CL lesions caused by various species of *Leishmania*. Recently, an AI-based system was developed to detect *Leishmania* parasites in microscopic images [62]. Therefore, we believe that AI can enhance the diagnosis of CL by processing macroscopic and microscopic images of skin lesions and parasites, respectively. These innovations could improve the parasitological diagnosis of CL, which remains the gold standard due to its high specificity. Furthermore, in our work, we built a DL model using pictures as the unit of analysis and, therefore, our results may not be extrapolated to conclude the diagnosis. To resolve this limitation, we will carry out a cohort study that uses the model constructed here to evaluate the accuracy of this instrument in real life. Furthermore, variations in image acquisition and quality, such as zoom, focus, lighting, and the presence of hair, pose challenges to the implementation of AI in clinical settings. The quality and field of view of the skin lesion image can have a significant impact on accuracy, and performance tends to improve in the absence of hair [55]. To overcome this challenge, we can include high- and low-quality images in our database. We use up to 10 pictures of each patient lesion, which poses a problem with non-independent data. We have two approaches to overcome this limitation: (1) increase the image database to use only one image of each lesion, or (2) apply generalized mixed models [37,63] to deal with problems of non-independence and convergence in the image database. Although manual image segmentation (cropping) was used in the model presented here, automating the segmentation and resizing processes are essential to create an application that health care professionals can easily use. This automation will be implemented in the ongoing study cohort.

## 5. Conclusions

Our study found that AlexNet has an accuracy of 95% in identifying CL lesions, which is a highly significant value for this complex diagnosis. Therefore, automated CL identification through AlexNet has the potential to assist clinicians in diagnosing skin lesions. These findings contribute to the development of a mobile application to support in the diagnosis of CL in health care services. Finally, our study represents an advancement in the application of DL as a new diagnostic tool in CL, with potential for application in the diagnosis of other infectious skin diseases.

The automated diagnosis of CL lesions using AI can help health care professionals by improving the accuracy of clinical assessments in real time. Therefore, AI can be a valuable tool to benefit health services by facilitating a differential diagnosis of skin lesions. In settings where professionals are not directly involved in the development of ML techniques, trained technicians can form a team to assist experts in creating automated diagnostic tools. This approach combines clinical research focused on diagnosis, treatment, and prevention with technological research involving AI and mobile application development. This collaborative effort has the potential to contribute to the early diagnosis and timely treatment of CL.

## Figures and Tables

**Figure 1 biomedicines-12-00012-f001:**
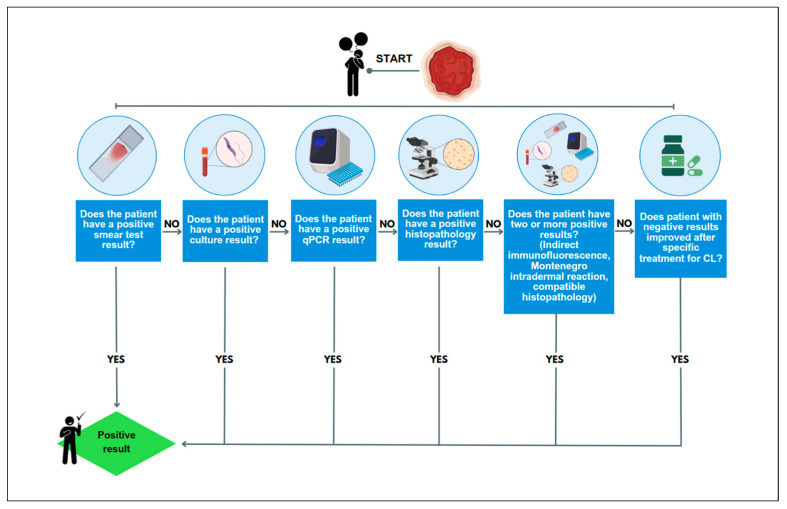
Patient flow at Leishmaniasis Clinic at the University Hospital of Brasilia. Source: created using BioRender.com. Accessed on 19 October 2023.

**Figure 2 biomedicines-12-00012-f002:**
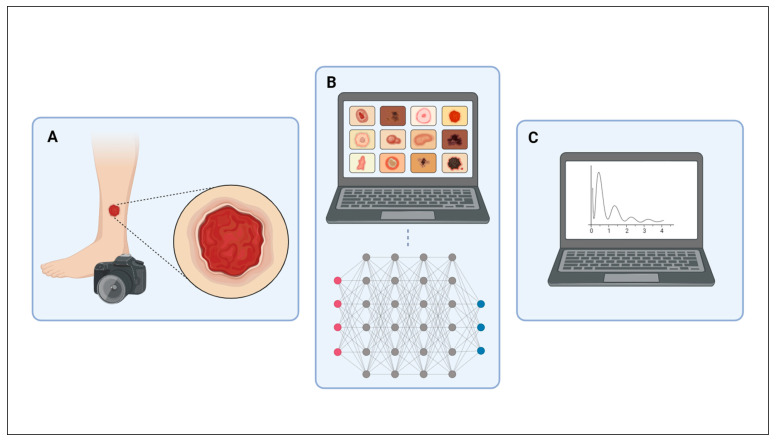
Scheme summarizing research methods. (**A**) Pictures of skin lesions; (**B**) sorting into folders, resizing pictures and simulations; (**C**) analysis of AlexNet’s performance. Source: created using BioRender.com. Accessed on 19 October 2023.

**Figure 3 biomedicines-12-00012-f003:**
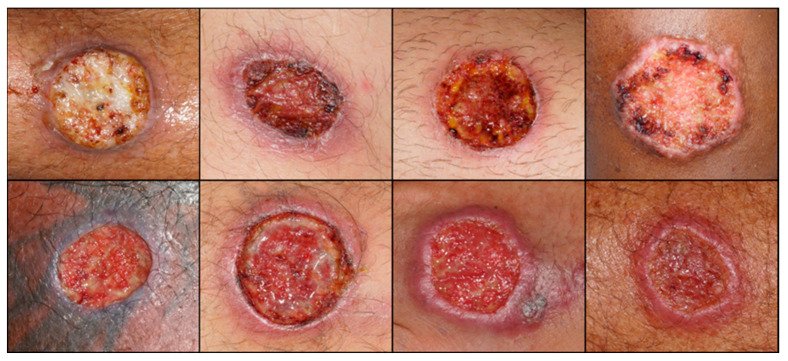
Examples of leishmaniasis skin lesions used in this study.

**Figure 4 biomedicines-12-00012-f004:**
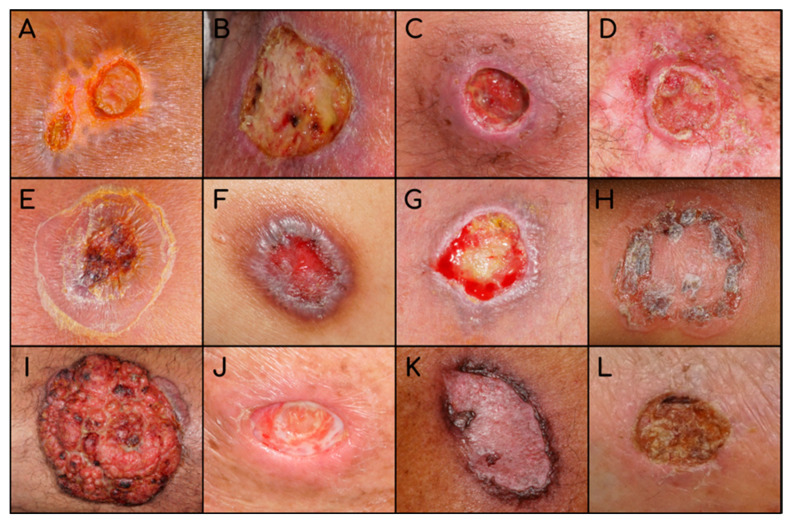
Examples of other types of skin lesions. (**A**) ecthyma; (**B**) erysipelas; (**C**) pyoderma; (**D**) tertiary syphilis; (**E**) sporotrichosis; (**F**) lichen planus; (**G**) lichen simplex chronicus; (**H**) tinea corporis; (**I**) SCC; (**J**) actinic keratosis; (**K**) porokeratosis; (**L**) venous insufficiency.

**Figure 5 biomedicines-12-00012-f005:**
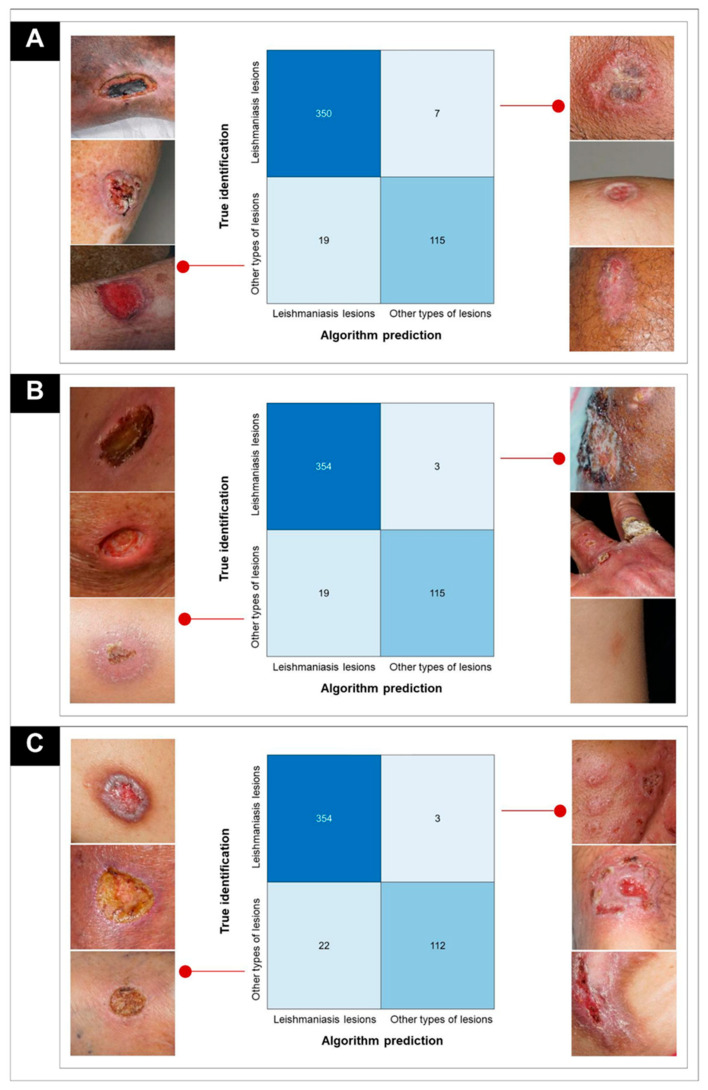
Confusion matrices of the simulations (cutaneous leishmaniasis lesions × lesions from other dermatoses) showing some of the CL pictures that were erroneously identified as other types of lesions using AlexNet and some pictures of other types of lesions that were erroneously identified as CL using AlexNet. (**A**) Simulation 1; (**B**) simulation 2; (**C**) simulation 3.

**Figure 6 biomedicines-12-00012-f006:**
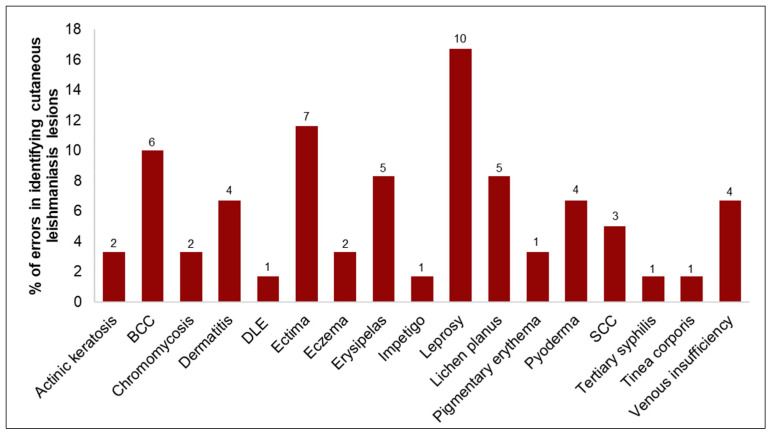
Percentage of dermatosis pictures incorrectly identified using the AlexNet network. Above each bar, there is the exact number of identifications as CL. BCC, DLE and SCC according to Table 1.

**Figure 7 biomedicines-12-00012-f007:**
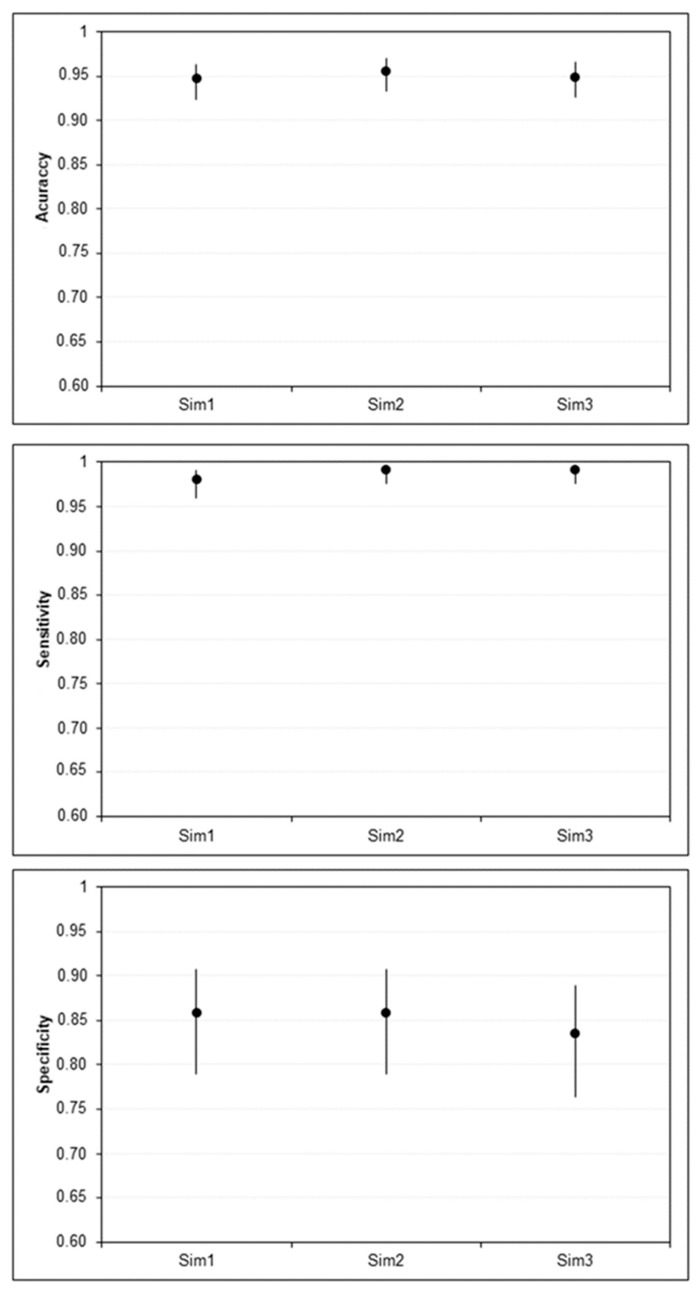
Accuracy, sensitivity, and specificity of the AlexNet network for identifying pictures of CL lesions with 95% Wilson confidence intervals (bars). Sim = simulation.

**Table 1 biomedicines-12-00012-t001:** Number of pictures of skin lesions related to the classification group, according to etiology.

Group	Disease	*n*
Protozoosis	Cutaneous Leishmaniasis	1787
Bacterioses	Ecthyma	55
	Erysipelas	24
	Impetigo	26
	Leprosy	57
	Pyodermite	98
	Tertiary syphilis	12
	Subtotal	272
Fungal	Chromomycosis	56
	Lichen planus	18
	Lichen simplex chronicus	12
	Paracoccidioidomycosis	7
	Sporotrichosis	22
	Tinea corporis	18
	Subtotal	133
Non-infectious	Actinic keratosis	8
	Basal cell carcinoma (BCC)	31
	Dermatitis	12
	Discoid lupus erythematosus (DLE)	16
	Eczema	15
	Fixed pigmentary erythema	5
	Livedoid vasculitis	12
	Lymphocytoma cutis	8
	Lymphoma	9
	Porokeratosis	7
	Psoriasis	2
	Rosacea	3
	Squamous cell carcinoma (SCC)	40
	Venous insufficiency	98
	Subtotal	266
Grand total		2458

## Data Availability

The data presented in this study are available on request from the corresponding author. The data are not publicly available due to ethical restrictions.

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
