# Peer review of "Automated Identification of Cutaneous Leishmaniasis Lesions Using Deep-Learning-Based Artificial Intelligence"

_biomedicines, 2023, doi:10.3390/biomedicines12010012_

Round 1
Reviewer 1 Report
Comments and Suggestions for Authors
The authors report a pleasant attempt to adapt artificial intelligence, currently very fashionable, with the clinical diagnosis of leishmania. Indeed diagnosis of cutaneous leishmaniasis is tricky and also for expert clinicians prompt diagnosis can be a challenge. Looking at the clinical pictures of leishmania and other diagnoses, I see that only ulcerative lesions have been considered, nevertheless CL can present also with different clinical presentation. The report is very interesting and AlexNet seem to be very sensitive to recognize diseases. My only suggestion is a historical paragraph about leishmaniasis: Nazzaro G, et al. Leishmaniasis: a disease with many names. JAMA Dermatol. 2014 Nov;150(11):1204
Author Response
We thank the Reviewer for the positive comments. We accepted the suggestion to include some historical aspects about leishmaniasis (lines 33-41) and we also cited the reference of Nazzarro et al (2014). For this reason, we modified the numbers of all references in the manuscript.
Reviewer 2 Report
Comments and Suggestions for Authors
The work presented by the authors is interesting. I have only some minor issues:
- in the methods, why did you perform three simulations? Please explain more clearly the rationale of this choice
- why do you think some specific dermatoses such as leprosy and BCC were more often confused by the algorithm? This point is not enough explored in the discussion
- you compare with some studies on application of algorithms to several skin pathology, but what about comparison and discussion with algorithms used to identify specifically algorithms? Some papers can be find in a recent review on the topic (DOI 10.1016/j.prp.2023.154362) which you may consider to read and quote
Author Response
The work presented by the authors is interesting. I have only some minor issues:
We thank the Reviewer for the positive comments.
- in the methods, why did you perform three simulations? Please explain more clearly the rationale of this choice
We conducted three simulations to assess variance and enhance the reliability of the findings. The outcomes of the simulations enable detection of errors or inconsistencies. Furthermore, the repetition of simulations mitigates bias linked to possible external factors that may influence the results if performed only once. The three simulations enable the calculation of the mean and descriptive statistics, allowing estimation of result uncertainties. It is crucial to communicate data reliability in the article.
- why do you think some specific dermatoses such as leprosy and BCC were more often confused by the algorithm? This point is not enough explored in the discussion
We explored better the results of confusion matrices regarding other dermatoses such as BCC and leprosy in the discussion section (lines 371-377). The presence of various diseases in the "other dermatoses" category results in few images of each disease type, which can complicate the training of the AI algorithm for leprosy, ecthyma, and BCC, increasing the likelihood that a particular disease will have a high false-positive rate.
- you compare with some studies on application of algorithms to several skin pathology, but what about comparison and discussion with algorithms used to identify specifically algorithms? Some papers can be find in a recent review on the topic (DOI 10.1016/j.prp.2023.154362) which you may consider to read and quote
In the discussion section we show the accuracy results of various algorithms used to identify skin lesions (lines 324-352). We accepted the reviewer's suggestion to include the review by Marletta et al. (2023)